# The Putative Role of Neuroinflammation in the Interaction between Traumatic Brain Injuries, Sleep, Pain and Other Neuropsychiatric Outcomes: A State-of-the-Art Review

**DOI:** 10.3390/jcm12051793

**Published:** 2023-02-23

**Authors:** Alberto Herrero Babiloni, Andrée-Ann Baril, Camille Charlebois-Plante, Marianne Jodoin, Erlan Sanchez, Liesbet De Baets, Caroline Arbour, Gilles J. Lavigne, Nadia Gosselin, Louis De Beaumont

**Affiliations:** 1Division of Experimental Medicine, McGill University, Montreal, QC H3A 0C7, Canada; 2CIUSSS-NIM, Hôpital du Sacré-Coeur de Montréal, Montreal, QC H4J 1C5, Canada; 3Douglas Mental Health University Institute, Montreal, QC H4H 1R3, Canada; 4Faculty of Medicine and Health Sciences, McGill University, Montreal, QC H3G 2M1, Canada; 5Department of Psychology, University of Montreal, Montreal, QC H3T 1J4, Canada; 6Hurvitz Brain Sciences Program, Sunnybrook Research Institute, Toronto, ON M4N 3M5, Canada; 7Pain in Motion Research Group (PAIN), Department of Physiotherapy, Human Faculty of Medicine, University of Montreal, Montreal, QC H3T 1C5, Canada; 8Physiology and Anatomy, Faculty of Physical Education & Physiotherapy, Vrije Universiteit Brussel, 1050 Brussel, Belgium; 9Faculty of Nursing, Université de Montréal, Montreal, QC H3T 1J4, Canada; 10Faculty of Dental Medicine, University of Montreal, Montreal, QC H3T 1C5, Canada; 11Department of Surgery, University of Montreal, Montreal, QC H3T 1J4, Canada

**Keywords:** traumatic brain injury, headache, concussion, neuroinflammation, microglia, sleep, pain, Alzheimer’s, dementia

## Abstract

Sleep disturbances are widely prevalent following a traumatic brain injury (TBI) and have the potential to contribute to numerous post-traumatic physiological, psychological, and cognitive difficulties developing chronically, including chronic pain. An important pathophysiological mechanism involved in the recovery of TBI is neuroinflammation, which leads to many downstream consequences. While neuroinflammation is a process that can be both beneficial and detrimental to individuals’ recovery after sustaining a TBI, recent evidence suggests that neuroinflammation may worsen outcomes in traumatically injured patients, as well as exacerbate the deleterious consequences of sleep disturbances. Additionally, a bidirectional relationship between neuroinflammation and sleep has been described, where neuroinflammation plays a role in sleep regulation and, in turn, poor sleep promotes neuroinflammation. Given the complexity of this interplay, this review aims to clarify the role of neuroinflammation in the relationship between sleep and TBI, with an emphasis on long-term outcomes such as pain, mood disorders, cognitive dysfunctions, and elevated risk of Alzheimer’s disease and dementia. In addition, some management strategies and novel treatment targeting sleep and neuroinflammation will be discussed in order to establish an effective approach to mitigate long-term outcomes after TBI.

## 1. Introduction

It has been reported that 69 million (95% CI 64–74 million) individuals are estimated to suffer traumatic brain injuries (TBI) from all causes each year, and that such injuries are associated with the development of consequences that may persist for years after the injury, such as pain, psychiatric, neurological, motor, and neurobehavioral issues, as well as with an increased risk of neurodegeneration [1,2,3,4]. Among the possible underlying pathological mechanisms of these consequences is neuroinflammation, which is an inflammatory response within the central nervous system (CNS) thought to be mediated by the production of cytokines, chemokines, reactive oxygen species (ROS) and other secondary messengers [5]. Although neuroinflammation is considered an adaptive and essential response following acquired traumatic injuries, edema, demyelination, and cellular and axonal damage were found to be associated with excessive neuroinflammation in chronic TBI [6]. Furthermore, evidence suggests that neuroinflammation secondary to acquired traumatic injuries, such as TBI, could play a central role in the development of chronic pain and also several tauopathies [7,8], such as Alzheimer’s disease, Parkinson’s disease, and chronic traumatic encephalopathy [6,9]. In this context, whether neuroinflammation could potentially precipitate or even cause neurodegenerative processes requires special attention.

In parallel, an important factor associated with TBI is disrupted sleep [10,11,12,13,14,15,16,17]. A recent meta-analysis showed that 50% of individuals with TBI report sleep disturbances, whereas approximately one-third of these patients report a sleep disorder [14]. Some of the sleep disturbances reported by those individuals include increased need for sleep even up to 6 months following the injury, obstructive sleep apnea, insomnia, narcolepsy-like symptoms, excessive daytime sleepiness, fatigue, and circadian-rhythm disturbances [16]. The causes of sleep disturbances following a TBI vary considerably: some might occur because of changes in patients’ lifestyle, comorbidities caused by the trauma such as pain and mood changes, or CNS structural damage and pathophysiological mechanisms incurred following a TBI [11]. In addition, one cannot exclude the possibility that premorbid sleep disturbances were exacerbated by TBI. Given the known deleterious consequences of sleep disturbances on alertness, concentration, and vigilance, individuals experiencing sleep disturbances following traumatic injuries are also more prone to subsequent traumatic injuries [18]. Sleep disturbances are also associated with poor prognosis following TBI [14,16,19] as well as declining overall health [20]. Understanding the complex interaction between sleep and TBI is not only essential if we aim to design and implement new management strategies, but it may also be instrumental in understanding their potential mediating role on the development of TBI-prone chronic pain diseases and neuropsychiatric conditions. In that context, and based on emerging research, there is reason to believe that neuroinflammation could entail a promising mechanistic linkage underlying poor prognosis in TBI patients experiencing sleep disturbances [21,22].

In this state-of-the-art review, the objective is to explore the putative role of neuroinflammation from fundamental and clinical perspectives into the relationship between sleep and TBI, in particular to their complex interplay with TBI-prone chronic pain and other neuropsychiatric outcomes. For that purpose, the information is organized and grouped into three main sections: “State-of-the-art overview of mechanisms between TBI, neuroinflammation, and sleep”, where a global perspective on mechanisms and outcomes (including chronic pain) of the multifaceted relationship between these conditions is described; “Neuroinflammation and other neuropsychiatric outcomes in the context of sleep disturbances and TBI”, namely mood disorders, cognitive dysfunctions and neurodegeneration; and “Future directions for clinical practice: targeting neuroinflammation”, which include sleep and neuroinflammation specific possible management options.

## 2. State-of-the-Art Overview of Mechanisms between TBI, Neuroinflammation, and Sleep

### 2.1. Mechanisms of Neuroinflammation Following TBI

Following tissue or nerve injury caused by a fracture or a TBI, a series of reactions generated by the body are triggered to allow the rapid return to homeostasis [23,24]. This inflammatory process can, in some cases, lead to an excessive and prolonged immune response, thus triggering a complex cascade of events such as chronic inflammation of the CNS (i.e., neuroinflammation) [25,26,27].

In acute and subacute neuroinflammation, microglia, which are the resident macrophages of the CNS, actively monitor the brain microenvironment and react when they encounter various elements of the CNS such as injured cells and pathogens, following TBI [5,28]. In response, microglia become activated and release cytokines (IL-1β, IL-6, TNF-a), nitric oxide (NO), and ROS, which are considered proinflammatory mediators [29]. This release leads to an acute neuroinflammatory response that is postulated to be beneficial to the CNS, in order to clear cellular debris via phagocytosis [28]. The resolution of the neuroinflammatory process is mediated by anti-inflammatory cytokines and the release of anti-inflammatory lipid mediators such as lipoxins, resolvins and neuroprotectins [28]. However, a neuroinflammatory response that persists over time may be detrimental and ultimately lead to neuronal death [30].

A growing body of evidence suggests that microglial activation is associated with synaptic dysfunction/dysregulation, mostly by altering long-term potentiation (LTP) [27]. These LTP, in turn, affect cognitive function, in particular long-term memory [9]. Synaptic dysfunction was also found to precede neuronal pathology such as tauopathies [31], and thus, such that neuroinflammation could be involved in both the onset and the progression of neurodegenerative diseases [6]. For instance, recent animal studies showed that rather than simply activating microglia, neuroinflammation can induce an exaggerated microglial response within the CNS thus “priming”, the inflammatory system for an increased vulnerability to a “second hit”, consequently favoring subsequent neuropsychiatric and neurodegenerative complications [32,33]. This inappropriate neuroinflammatory response activates several self-propagating cycles, causing apoptosis, synaptic dysfunction, impaired regeneration and the production of amyloid-beta (Aβ) and phosphorylated tau, thereby exacerbating behavioral and cognitive impairments [32].

### 2.2. TBI and Neuroinflammation

Brain damage related to the mechanical force applied to the brain in TBI is referred to as the primary insult (i.e., skull fractures, intracranial hematoma, lacerations, and contusions, diffuse axonal injury) [34]. The secondary insult refers to ischemia caused by various mechanisms, including intracranial hypertension, that compromise the balance between oxygen delivery to neurons and cerebral oxygen consumption [34]. It generates complex and interrelated neurochemical changes, including an extracellular increase in excitatory amino acids, ROS production, increased intracellular sodium and calcium concentration, mitochondrial dysfunction, and a long-lasting inflammatory response that may ultimately lead to cell death [34,35,36]. The significant activation of microglia, as part of the secondary insult, leads to the release of cytokines responsible for neuroprotection (anti-inflammatory process) and neurodestruction (toxic and pro-inflammatory process) [33,37,38]. The balance between the neuroprotective and neurodestructive components is precarious. When a misalignment between these two components occurs to the advantage of the latter, there is an increased risk of progressive brain damage that may persist and progress into chronic neurodegeneration [39]. Indeed, recent studies have found an increased levels of proteins involved in pathological processes of neurodegeneration, such as α-synuclein, Aβ, and tau in TBI patients, which play a major role in the development of neurodegenerative diseases such as Parkinson’s and Alzheimer’s diseases [40,41,42,43]. Given the high prevalence of TBI across the lifespan, understanding the complex interaction between neuroinflammation and its associated neurodegenerative processes is essential to improve patients’ clinical outcomes.

Regarding polytrauma (i.e., the simultaneous traumatic injury of several regions of the body), numerous studies have shown a high incidence of TBI in individuals who have suffered orthopedic trauma, which is not surprising considering that they both share similar causative events (accidental falls, motor vehicle accidents, and accidents in a recreational setting) [44,45]. Therefore, the anatomical proximity of the upper extremities to the head is such that these two types of acquired traumatic injuries inevitably share somewhat comparable biomechanical characteristics [46]. Consequently, the occurrence of a polytrauma brings its share of challenges due to the overlapping pathophysiological mechanisms common to both injuries and their possible interactions. Indeed, the permeability of the blood–brain barrier (BBB) following a TBI facilitates peripheral factors to invade the CNS [47]. Thus, pro-inflammatory cytokines, such as IL-6, IL-1β, and TNF-α, released following a peripheral lesion lead to a significant increase in systemic inflammation [24,35]. Therefore, patients with a TBI and a concomitant peripheral injury are potentially at greater risk for an exacerbation of the ongoing neuroinflammatory response than patients with an isolated TBI or peripheral injury [48]. The latter becomes specially relevant in the context of chronic pain development, as the risk of neural sensitization is increased by the possibility of both peripheral and central neuroinflammation, which may be driving the onset of post-traumatic conditions such as post-traumatic headache and complex regional pain syndrome [49,50,51].

Importantly, attention needs to be directed toward pediatric populations as well, as childhood and adolescence is a time of elevated risk for TBI [52], and prognosis and treatment responses may differ from adults due to the neuronal and brain network developmental status [53,54]. For instance, TBI can influence hippocampal neuro-genesis, which can increase the risk of developing adult neurological and neurodegenerative diseases [55,56]. A recent large-scale study using diffusion-weighted imaging showed that children with persistent post-concussive symptoms (more than 6 months following mild TBI) had more white matter microstructural changes than those with less persistent symptoms or mild orthopedic injury, suggesting more neuroinflammation and axonal swelling [57]. Hence, future research on this population is encouraged, as one might suspect augmented risk for neurocognitive alterations, especially in cases where neuroinflammation persists.

### 2.3. TBI and Sleep

Sleep–wake disturbances, including excessive daytime sleepiness, fatigue, and insomnia, are frequently reported by patients with TBI [58], and actually patients with TBI are at higher risk of developing chronic sleep–wake disturbances [11,59]. These disturbances can occur immediately after the TBI and tend to persist over time, as longitudinal studies have reported these symptoms are presents 6 months, 12 months, and even 3 years after the TBI [11,60,61,62,63]. For instance, a meta-analysis in patients with chronic TBI (>6 months post-injury) showed that moderate-severe TBI was associated with elevated slow wave sleep (SWS), reduced stage 2, and reduced sleep efficiency [15], and a recent retrospective cohort study in war veterans, with a median follow-up rate of 8.4 years, showed that TBI was associated with insomnia at follow-up when compared with patients without TBI (hazard ratio = 2.07; 95%) [64]. Indeed, insomnia following mild TBI seems to be common and perhaps among the main causes of disability in these patients [65,66], and relevant factors such as female sex, black race, history of psychiatric illness, and intracranial injuries seem to lead towards different insomnia trajectories [67]. Another sleep-related consequence of TBI is increased sleepiness, especially in early stages, and research has highlighted the damage of orexin/hypocretin neurons, whose activation involve wakefulness, as a possible contributor of the association between this association [68]. For example, a study revealed that TBI patients in the acute stage of severe TBI showed increased sleep duration and earlier sleep onset, perhaps suggesting that in the short-term the injured brain enhances sleep need and/or decreases the ability to maintain wakefulness [69]. Importantly, mood disorders (e.g., anxiety and depression), which are also associated with neuroinflammation, have been suggested as potential mediators of this association, and as with many other chronic conditions, it is very difficult to disentangle their role and their respective contribution in sleep–wake disorders following TBI [58,70].

### 2.4. Inflammation and Neuroinflammation Regulates Sleep

In the healthy brain, experimental studies have shown that inflammatory levels, whether peripheral or within the CNS, affect sleep regulation [71,72]. Cytokines are thought to be among the main effectors linking sleep and inflammation: in fact, IL-1β and TNF-α are sleep regulatory cytokines known to promote longer and deeper sleep. Overexpression of IL-1β and TNF-α following TBI is therefore thought to at least partially contribute to the heightened need for sleep following an injury [73]. Although their effects are smaller, other cytokines and prostaglandins also display sleep regulatory properties [71]. In the daytime, inflammatory levels are associated with fatigue and sleepiness [74]. However, some of these effects seem to be level dependent, where an inverse relation can be observed at higher levels, with high pro-inflammatory cytokines levels being associated with disrupted and fragmented sleep [75]. This observation might partly explain why many chronic inflammatory diseases are associated with sleep disturbances [71]. For instance, in a population with high inflammation and depression, the administration of a TNF blockade significantly improved sleep consolidation [76].

Recent studies show that microglia could play a key mediating role on sleep regulation through their production of sleep regulatory cytokines [77]. Moreover, microglia morphology, phagocytosis activity, and their gene expression were also shown to follow circadian variations [78,79]. Therefore, the normal circadian release of cytokines might contribute to sleep regulation. However, in a study that administered minocycline to attenuate microglial activation in mice that underwent sleep deprivation, a suppression of the normal increase in sleep depth was observed, which did not seem to be mediated by changes in cytokines transcription [80]. These findings suggest that microglial activation play a role in sleep regulation following acute sleep deprivation. Interestingly, animal research has suggested that the duration of post-traumatic sleep is a period that may define vulnerability for a repeated brain injury, which could be more related to glial activation rather than orexin neurons damage [81]. Although it remains unclear as to how exactly neuroinflammation regulates sleep, particularly in clinical populations, current hypotheses include the modulation of synaptic transmission affecting sleep [82,83], and damage to sleep regulatory structures in the brain, such as the thalamus, pituitary, hypothalamus, and brainstem, leading to sleep disturbances and disorders [84]. Alternatively, activated microglia might affect sleep–wake cycles through alterations of hypothalamic neurons that produces hypocretin [77], leading to narcolepsy-like symptoms such as excessive sleepiness, heightened sleep propensity, and disrupted nocturnal sleep. Taken together, current evidence suggest that microglial function regulates sleep, identifying neuroinflammatory processes as potential causes of sleep disturbances in TBI [85]. Furthermore, it seems that the characterization of sleep after TBI is essential to understand better the development of different neuropsychiatric outcomes [86,87].

### 2.5. Sleep Affects Inflammatory and Neuroinflammatory Processes

Sleep occupies approximately one-third of our lives and plays a central role in maintaining physiological homeostasis. Sleep is also crucial to TBI recovery as it is involved in metabolic and autonomic regulation [72,88,89,90], synaptic plasticity [91,92], memory consolidation [93] and other cognitive functions [94,95], mood regulation [96], as well as glymphatic clearance of metabolites from the brain [97,98]. In addition, sleep is an important regulator of the immune system [71,72]. It comes as no surprise that disturbed sleep has been shown to affect inflammation [71,72,99]. However, the inflammatory response to sleep loss can change depending on the chronicity: acute sleep deprivation results in lower IL-6, IL-1β, and TNF-α levels, whereas prolonged sleep restriction leads to elevated cytokines levels and increased inflammatory gene expression [71]. Sleep disturbances as well as short sleep duration have been associated with elevated inflammatory markers [100]. In patients with insomnia, the presence of short sleep, sleep fragmentation, and reduced slow-wave sleep were associated with higher inflammasome levels [101]. Sleep loss has also been shown to impact neuroinflammation and microglia. In many animal models, acute and chronic sleep loss generally affects microglial morphology, gene expression, activation [78]. After both chronic sleep loss and/or restriction in mice, microglial activation as well as microglial and astrocytic phagocytosis of synaptic components were observed, which may be a response to higher synaptic activity associated with prolonged wakefulness [102]. The authors suggested that sleep loss promotes “housekeeping” of heavily used synapses to downscale them, but these processes might also result in enhanced susceptibility to brain damage. Furthermore, it has been postulated that stress and poor sleep can trigger glial overactivation and a subsequent low-grade neuroinflammatory state, characterized by high levels of IL-1β and TNF-α, which, in turn, increases the excitability of CNS neurons through mechanisms such as long-term potentiation and increased synaptic efficiency [103].

Peripheral inflammation can also lead to neuroinflammation in the context of poor sleep [104,105]. Chronic sleep loss and sleep disorders such as obstructive sleep apnea have been associated with compromised BBB [106,107], and could result in an increased invasion of peripheral immune cells and cytokines into the CNS, thus contributing to neuroinflammation. One study used a 3-day sleep deprivation protocol in rats, and observed that sleep loss was associated with a cascade of pathological mechanisms, including exacerbated cortisol levels suggestive of a hypothalamic–pituitary–adrenal (HPA) response, altered circadian oscillations of clock genes expression, disrupted BBB integrity and microglial activation with elevated pro-inflammatory cytokines levels (IL-6, IL-1β, and TNF-α) [108].

Overall, sleep disturbances may contribute to poor health outcomes partly through detrimental chronic inflammation that can perpetuate tissue damage [84], which could exacerbate TBI-related inflammation and neuronal damage. Taken together, these recent findings highlight the bidirectional relationship between sleep–wake cycles and neuroinflammation. In the context of TBI, we hypothesize that the occurrence of sleep disturbances could be caused in part by neuroinflammatory processes following the trauma, which then, in turn, could synergically promotes neuroinflammatory-related tissue damage.

### 2.6. Neuroinflammation and Chronic Pain in the Context of Sleep Disturbances and TBI Chronic Pain

A common consequence of both TBI and poor sleep is chronic pain (i.e., pain lasting longer than 6 months). The interaction between sleep and pain problems is complex and likely bidirectional [109], and the most common pain condition after TBI appears in the form of headache, nowadays named persistent headache attributed to traumatic injury to the head [110,111]. The prevalence of persistent post-traumatic headache is as high as 57.8% (95% confidence interval [CI], 55.5–60.2%) across different time points [112]. In addition to headache, the onset of pain after TBI has also been reported in the neck, in the shoulders, or in the upper limbs [113]. In fact, TBI is accompanied by another pain diagnosis in more than 40% of cases [45], and in moderate-to-severe TBI, musculoskeletal complaints (stiffness and aching in joints) are present in 79% of patients assessed, more than 15 years after trauma [114]. While different potential underlying mechanisms have been identified, a possible underlying mechanism to the sleep and pain interaction relates to inflammatory processes (low grade inflammation or neuroinflammation) [71,115,116,117]. Accordingly, a recent study highlighted the role of IL-6 in the development of pathological pain, whose receptors seemed to be elevated in the spinal cord and nerve root ganglia in chronic pain states [118]. In addition, prostaglandins, other cytokines such as IL-1 as well as TNF are considered important pronociceptive factors that could mediate the association between sleep loss and increased pain in the context a chronic pain condition, such as post-traumatic headache. Moreover, it seems that melatonin, an endogenous substance produced in the pineal gland that is mainly associated with sleep–wake circadian regulation, is also linked with suppressing pain and inflammation [119]. Indeed, low melatonin has been postulated as a potential moderator for the association between chronic pain, sleep architecture, and immunometabolic traffic, as it can downregulate inflammatory mediators including prostaglandins and cytokines [119]. A recent review also highlighted the lower levels of melatonin on neuroinflammation and oxidative stress resulting from TBI [120], and a pre-clinical study among severe TBI patients found lower serum melatonin levels in the surviving patients [121]. It has also been shown that individuals with pain and mild TBI may need more time to sleep, and the authors concluded that pain could be associated with more pronounced sleep need in these individuals [122].

Additionally, yet not specifically related to trauma populations, other clinical studies have also found peripheral deficiencies compatible with neuroinflammation in pain syndromes such as fibromyalgia [123], where sleep disturbances are present in most of the cases [103]. Hence, the activation of microglia and astrocytes seems to be critical in the development of most chronic pain conditions [103,124].

## 3. Neuroinflammation and Other Neuropsychiatric Outcomes in the Context of Sleep Disturbances and TBI

### 3.1. Mood

TBI, sleep and pain are all major risk factors for mental health disorders such as anxiety or depression [125,126]. Interestingly, these frequent long-term consequences of traumatic injuries share neuroinflammation among key pathophysiological mechanisms [58,127,128,129]. According to a recent systematic review [130], the presence of depressive and/or anxiety symptoms in TBI samples was found to be associated with higher concentrations of serum and CSF, CRP, CSF-derived markers of sVCAM-1, sICAM-1, and sFAS, and IL-10, IL-8, IL-6, and TNF-α. Acute measures of some of these biomarkers predicted the onset of depression at 6 and 12 months post-injury.

Robust animal evidence has linked sleep deprivation to depression and anxiety-like behaviors partly through neuroinflammatory processes [21,108,131,132]. Following sleep deprivation in mice that underwent a TBI, lower corticosterone, enhanced neuroinflammation, exacerbated evidence of neuronal injury, and anxiety-like behaviors were observed as compared to brain-injured mice without sleep deprivation [21].

Taken together, traumatic injuries seem to interact with sleep disturbances in the installation of persistent trauma-related sequelae affecting mood, potentially through shared neuroinflammatory processes.

### 3.2. Cognitive Dysfunctions and Neurodegeneration

Neuroinflammation is now recognized as a key pathological mechanism to cognitive aging, neurodegeneration and Alzheimer’s disease [133]. Meta-analyses have concluded that both sleep disturbances and TBI are risk factors for cognitive decline and incident dementia [7,134,135]. There is increasing evidence suggesting that sleep disturbances interact and/or contributes to peripheral inflammation as well as neuroinflammation to predict cognitive dysfunctions and dementia risk. For instance, inflammatory levels have been shown to moderate the association between sleep disturbances and obstructive sleep apnea with dementia risk [136,137]. In animal models, sleep deprivation or fragmentation lead to cognitive dysfunctions and neurodegenerative processes, at least partly through its effect on neuroinflammation (microglial activation, cytokines production, complement activation) [104,132,138,139]. In mice, one group used a 2-month chronic sleep fragmentation protocol, which resulted in the activation of microglia, endosome-autophagosome-lysosome pathway dysfunction, cortical and hippocampal Aβ accumulation, spatial learning and memory impairments, and anxiety-like behaviors [132]. In sleep deprived rats, inhibiting microglial activation mitigated spatial memory impairments, reduced deleterious effects on neurogenesis and gliosis in the hippocampus, and promoted anti-inflammatory cytokines over pro-inflammatory cytokines [139], supporting the causal role of microglial activation in sleep deprivation-induced cognitive dysfunction. Alternatively, sleep disturbances can also directly promote neurodegenerative processes through other mechanisms, such as a lower metabolic clearance of Aβ via the glymphatic system [97]. Interestingly, convincing evidence indicates that neuroinflammation could effectively modulate neurogenesis at different stages, including proliferation, differentiation, migration, survival of newborn neurons, maturation, synaptogenesis, and neuritogenesis among others [140]. Finally, a recent TBI study concluded that post-injury sleep fragmentation engages the dysfunctional post-injury HPA axis, enhances inflammation, and compromises hippocampal function [141]. The latter study suggested that external stressors that disrupt sleep have an integral role in mediating outcome after brain injury. Thus, both systemic inflammation and neuroinflammation can alter adult hippocampal neurogenesis in neurodegenerative disorders. For a more detailed an extensive review in The Dialogue Between Neuroinflammation and Adult Neurogenesis, please see [140].

Following a TBI, accumulating evidence is showing that neuroinflammation contributes to initial neuronal damage and cognitive dysfunction, but also long-term cognitive impairments, neurodegeneration, and risk of developing dementia [8,84,142]. After a single TBI, patients show evidence of white matter degeneration and persistent neuroinflammation up to 18 years post-injury [143]. In a mouse model of TBI, the neuroinflammatory response was found to drive synaptic degeneration and cognitive decline, which was abolished by complement inhibition, suggesting causality [144]. In humans, concomitant tau aggregation and neuroinflammation was observed using neuroimaging in mild TBI patients [41]. Moreover, it has been reported that TBI can also augment the formation of amyloid-b plaques and tau neurofibrillary tangles (NFTs) through inflammation-dependent gene expression and transcription factor activation, which could, in turn, produce sleep disturbances. Importantly, NFTs are another crucial feature of Alzheimer’s disease [84].

Taken together, these findings suggest that the feedback positive loop between TBI, sleep disturbances and neuroinflammation can result in further cognitive dysfunctions and even neurodegeneration.

A summary of the abovementioned interactions and mechanisms can be observed in Figure 1.

## 4. Future Directions for Clinical Practice: Targeting Neuroinflammation

### 4.1. Sleep as a Therapeutic Target to Inhibit Neuroinflammation

Many sleep disturbances and disorders are treatable, often using non-pharmaceutical therapeutic strategies, thereby make it an appealing treatment target in order to reduce trauma-related neuroinflammation and improve patients’ lives [115]. Cognitive behavioral therapy (CBT), the gold-standard treatment for insomnia, has proven to be an efficient way of improving sleep quality and restoring inflammatory levels [145,146,147]. CBT has also proven effective in TBI patients and show promise for mitigating patients’ inflammation-related symptoms such as depression, anxiety and pain [148], along with other non-pharmaceutical strategies such as blue light therapy, problem solving treatment, and combined sleep hygiene interventions [149]. Moreover, in addition to sleep, CBT can also be directed towards pain and other associated disorders such as depression and anxiety in a hybrid approach. Hybrid CBT has shown promising results and it can also be carried out online to improve treatment compliance [150,151,152].

Although sleep medications, such as benzodiazepines, hypnotics, and sedating antidepressants, could help treat sleep disturbances, especially in the short term, and thereby have the potential to reduce neuroinflammation and TBI’s related consequences linked to poor sleep [153], this remains to be investigated thoroughly and cautiously. Although still controversial, usage of sleeping pills has been associated with elevated risk of incident dementia [154,155], Moreover, sleeping medications generally perform worse than behavioral techniques such as CBT in treating sleep disturbances in the long term [156].

Nonetheless, a sleep aid with interesting potential is melatonin, as it has been associated with the inhibition of excessive microglial activation [157]. In addition to its endogenous secretion at night promoting adequate sleep–wake cycles, melatonin is also available as a dietary supplement. Among the proposed mechanisms underlying melatonin’s downregulating action on microglial activation is through its role as an antioxidant, therefore reducing ROS [157]. It is also possible that melatonin supplementation could help reduce neuroinflammation through its effect on sleep regulation, although this remains to be confirmed. Interestingly, animal models showed that melatonin administration increased bone fracture healing [158], reduced neuroinflammation and promoted neuroprotection following a TBI [159,160]. In patients that sustained a TBI, a meta-analysis showed that melatonin has a positive effect on pathological findings, neurological status, neurobehavioral outcomes, and cognition [161]. However, it needs to be highlighted that the majority of the included studies were in animal models (i.e., 15 studies in animal models and two in human populations), and that the included human studies were considered to have low quality and were of uncertain significance. Furthermore, a recent randomized clinical trial in a pediatric population with mild TBI (n = 99) showed no significant difference in post-concussive symptoms between the use of melatonin at two different dosages (i.e., 3 and 10 mg) and placebo [162], yet a secondary analysis as per protocol of these data showed some improvements in sleep symptoms with melatonin [163]. Therefore, although there is evidence supporting the use of melatonin treatment after TBI to improve different behavioral and pathological outcomes based on animal models, data remain equivocal in human clinical populations.

### 4.2. Specifically Targeting Neuroinflammation to Improve Sleep and Trauma-Related Outcomes

Given its relevance and repercussions in different ambits of health, treating neuroinflammation emerges as a critical goal in the management of traumatically injured patients. Whereas peripheral neuroinflammation might be initially targeted with common anti-inflammatory medications, aiming to act on neuroinflammatory processes is much more complex, as some anti-inflammatories can disturb sleep as well, microglia appears as a primary treatment target for novel therapeutics aiming to tackle neuroinflammatory processes, including its selective abolition in animal models [164,165], or being targeted by nanoparticles [166]. Moreover, there are already several inhibitors of TNF-α and IL-1β that are available for clinical use, yet none of them are exempt of potentially serious side effects [167,168]. In addition, the use of psychedelics is receiving again a lot of attention in recent years due to its powerful properties to treat pain and mood disorders, as they have shown potential neuro-restorative effects and anti-neuroinflammatory and pro-immunomodulatory actions [169,170]. Indeed, the effects of some of these compounds is currently being studied in sleep as well [171]. Nonetheless, more development in this line of investigation is needed in the future. For a more detailed summary of pharmacological therapies on TBI, please refer to a recent review of phase 3 clinical trials on this population [172], which highlights key targets for future research.

An important non-pharmacological treatment option is exercise, given that exercise increases astrocytic activation, more specifically glial fibrillary acidic protein expression in hippocampal astrocytes in the stratum radiatum, a region that contains numerous astrocytes and is relevant for learning and memory [103,173]. Exercise is known to become anti-inflammatory or neuroprotective in several neuroinflammatory diseases. It is possible that exercise also reduces gliosis and glial proliferation [103]. Moreover, via its action on CNS glial cells, regular aerobic exercise has been shown to provide an adaptive advantage against perturbations to homeostasis, such as immunological challenge or ageing in animal models [174]. A systematic review and meta-analysis involving 13 RCTs and 514 participants, revealed that physical activity had positive effects on decreasing TNF-α and CRP (pro-inflammatory), while significantly improving BDNF and IGF-1 (neuroprotective) [175]. Furthermore, exercise is a great option for several sleep disorders including insomnia or sleep apnea [176,177], as it can regular cortisol, release endorphins, and decrease fat among others.

Non-pharmacological integrative approaches including mind/body therapies such as yoga, breathing exercises, meditation, all of them being associated with sleep quality improvement as well, have also been demonstrated to reduce pro-inflammatory cytokines and have proved some positive effects on depression, anxiety, cognition, and pain [178]. Moreover, several plant-based interventions (herbs/spices) currently under investigation [179]. While their non-invasiveness and harmless nature make them appealing as supportive therapy, more research is needed before obtaining any firm conclusion regarding their efficacy.

Other emerging techniques that can be used to target pain and sleep disorders, and specifically neuroinflammation, are non-invasive brain stimulation techniques, such as repetitive transcranial magnetic stimulation (rTMS) [180,181]. In different animal studies, rTMS reduced neuroinflammation by modulating astrocytes and microglia activity, reducing TNF-α, and increasing GABA, which can control excitotoxicity [182,183,184,185]. Additionally, clinical rTMS studies showed increases in serum GABA and BDNF in patients with chronic insomnia [186]. Thus, rTMS could be used not only to manage chronic pain patients but also to reduce their transition to chronicity by tackling the underlying neuroinflammatory mechanisms [115,185], which becomes especially relevant when applied to traumatically injured patients. While other techniques such as transcranial direct current stimulation, transcranial alternating current stimulation or vagal nerve stimulation hold potential in treating neuroinflammation [181,187], research is still lacking.

## 5. Conclusions

While the innate immune response following a TBI is necessary for recovery, its often prolonged and excessive nature contributes, paradoxically, to worsen outcomes. In that way, TBI leads to a state of peripheral/central neuroinflammation, which can be associated with sleep disturbances. Additionally, TBI and sleep disturbances also exacerbate the neuroinflammatory state, complicating these deleterious interactions even more, and potentially all leading to mood disorders, pain, cognitive deficits and neurodegeneration states. Importantly, finding treatment strategies, such as treating sleep disturbances or using non-invasive brain stimulation to reduce or modulate pro-inflammatory processes, can be useful in order to help TBI patients’ physiological, psychological and cognitive health. 

## Figures and Tables

**Figure 1 jcm-12-01793-f001:**
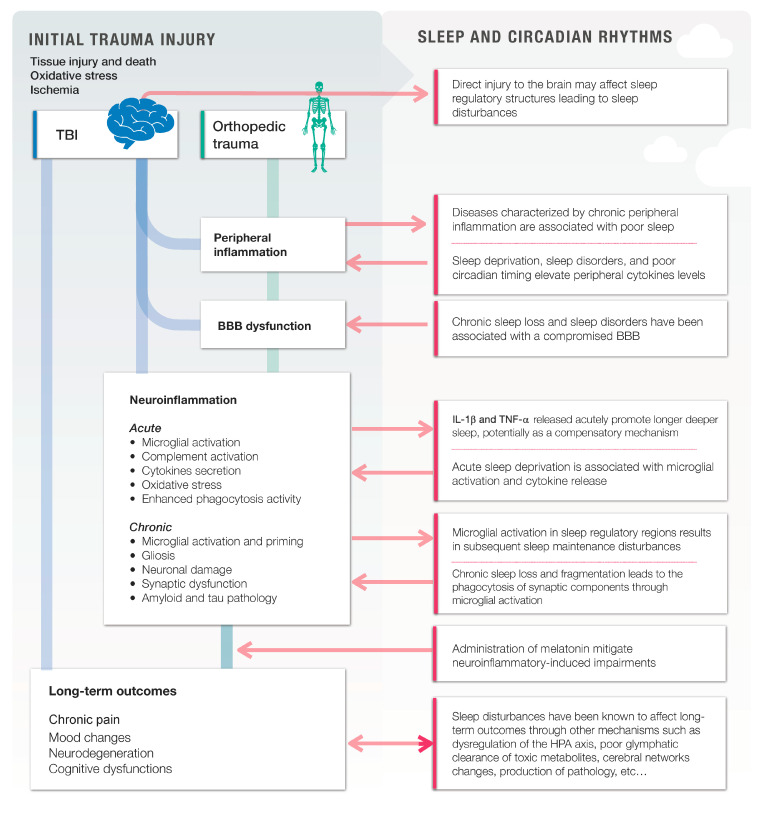
The putative role of neuroinflammation in the acquired traumatic injuries and related sleep disturbances. Traumatic brain injuries (TBI) and orthopedic traumas (OT) both lead to a downstream pathophysiological cascade that includes peripheral inflammation, blood–brain barrier (BBB) dysfunction, and neuroinflammation. In turn, chronic neuroinflammation plays a role into the development of poor long-term outcomes. At each step of the way, sleep and sleep disturbances interact bidirectionally with traumatic pathological mechanisms and neuroinflammation. Poorer sleep exacerbates peripheral inflammation, BBB dysfunction, neuroinflammation, and worsen long-term outcomes. On the other hand, the trauma itself, its comorbidities, or neuroinflammation affect sleep regulation, which leads to a positive feedback loop, where neuroinflammation and sleep interact together to affect long-term outcomes following an injury.

## Data Availability

Not applicable.

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
