# Peer review of "The Putative Role of Neuroinflammation in the Interaction between Traumatic Brain Injuries, Sleep, Pain and Other Neuropsychiatric Outcomes: A State-of-the-Art Review"

_jcm, 2023, doi:10.3390/jcm12051793_

Round 1

Reviewer 1 Report

This review provides a helpful overview of the role of neuroinflammation in the relationship between sleep and TBI.

I recommend that the authors make some edits to the information provided on pharmacological agents:

1) Paragraph beginning from line 383 regarding the effects of melatonin. Some useful theoretical information is provided on the mechanisms of melatonin here, however clinically relevant information is lacking. The authors state “In patients that sustained a TBI, a meta-analysis shows that melatonin has a positive effect on pathological findings, neurological status, neurobehavioral outcomes, and cognition”. However the citations that are provided do not support this – they are a study of hypnotic and dementia (citation 149), a study of cognitive behavioural therapy and insomnia (citation 151), and a systematic review and meta-analysis relevant to the topic. However, the systematic review by Barlow et al. that is cited by the authors included 15 animal studies and 2 clinical studies, and Barlow et al. concluded that the clinical studies were low quality and of uncertain significance. Moreover, Barlow and colleagues have more recently published an article reporting the findings of the PLAYGAME clinical trial (NCT01874847) of melatonin intervention in PCS. This concluded that the clinical trial did not support the use of melatonin for the treatment of paediatric PCS (PMID: 32217739). I recommend your paper reflect this information or remove the section highlighting melatonin.

2) The paragraph beginning line 395 regarding anti-inflammatory medications is useful. However, this is a detailed topic that the current paragraph does not adequately cover. I understand that this detail is outside the scope of the current paper, and it may be worth directing the reader to a comprehensive review of this topic. A suggested recent review that could be cited is PMID: 35644464, which provides summary of all Phase 3 clinical trials of pharmacological therapies of TBI including anti-inflammatory agents.

Minor edits:

Line 108 – the explanation of primed vs activated microglia is not clear from the current description

Line 201 – the ‘and’ is not required

Line 340 – comma not required

Reviewer 2 Report

Very good review on an under-explored subject: trauma-related neuroinflammation and sleep disorders. 

The review is well written and easy to read. The problematic is well stated. 

The authors scan the literature with the answers provided or not. 

At first glance, the article seems rather unorganized, but I am not sure that it could be better. 

I think it would be better to separate what we know about humans and animals with a table for example.

Would it be possible to make a paragraph on pediatric head trauma? 

Very nice figure summarized. 
